# MULTI-TASK LEARNING WITH GRADIENT COMMUNICATION

## ABSTRACT

In this paper, we describe a general framework to systematically analyze current neural models for multi-task learning, in which we find that existing models expect to disentangle features into different spaces while features learned in practice are still entangled in shared space, leaving potential hazards for other training or unseen tasks. We propose to alleviate this problem by incorporating a new inductive bias into the process of multi-task learning, that different tasks can communicate with each other not only by passing hidden variables but gradients explicitly. Experimentally, we evaluate proposed methods on three groups of tasks and two types of settings (IN-TASK and OUT-OF-TASK). Quantitative and qualitative results show their effectiveness.

## 1 INTRODUCTION

Multi-task learning has been established as an effective learning method in deep learning, driving state-of-the-art results to new levels in a number of tasks, ranging from computer vision (Misra et al., 2016; Zamir et al., 2018) to natural language processing (Collobert & Weston, 2008; Luong et al., 2015).

While various kinds of neural multi-task learning approaches have been proposed, a commonality is the two types of feature spaces they defined to store task-specific features and task-invariant features. And then interaction mechanisms among different feature spaces will be explored in diverse ways, such as hard sharing (Collobert & Weston, 2008) or soft sharing (Misra et al., 2016; Ruder12 et al., 2017) schemes , flat (Yang et al., 2016; Liu et al., 2016) or structural sharing schemes (Hashimoto et al., 2016; Søgaard & Goldberg, 2016).

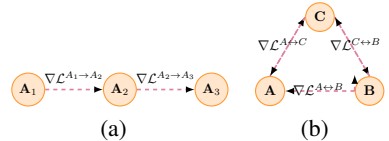

Figure 1: $A$, $B$, $C$ represent different tasks. $\nabla_\theta \mathcal{L}$ denotes the gradients with respect to $\theta$. (a). Gradient Passing within a task. (b). Gradient Passing across tasks.

Despite their great success, a potential limitation of most existing works is that shared space is entangled with private features since it is not enforced through the use of explicit loss functions (Bousmalis et al., 2016). Specifically, we would have extracted shared features and disentangled them into separated spaces. However, in reality, methods in use just collect all the features together into a common space, instead of learning shared rules across different tasks. This problem, which we define as "*pretend-to-share*", clearly suffers from the following issues: 1) IN-TASK SETTING: shared space is redundant and contaminated by task-specific features, leaving potential hazards for other training tasks. 2) OUT-OF-TASK SETTING: the learned model can not effectively adapt to new tasks since a real common structure is not discovered.

A few works explore this problem with adversarial training and shows that explicitly disentangling representations into private and shared spaces will improve model's performance (Bousmalis et al., 2016; Liu et al., 2017). In this paper, we propose an orthogonal technique to alleviate above problem. Specifically, we first describe a general framework for multi-task learning called **P**arameters **R**ead-**W**rite **N**etworks (PRaWNs) that can simply abstract the commonalities between existing typical neural models for multi-task learning. Then, we explain the reason for *pretend-to-share* problem within PRaWN that each task has the permission to modify shared parameters without any communication protocol, which makes it easier to update parameters along diverse directions.

Therefore, the problem is translated into this question: how to address the conflict among different tasks happening in the process of parameter updating. We resolve this conflict by introducing an inductive bias for multi-task learning. Specifically, different tasks can communicate with each other by sending (or receiving) both hidden variables and gradients. As shown in Fig.1(b), during the forward phase, tasks $A$, $B$, $C$ are capable of sending (and receiving) the gradients to (and from) each other. Then during the backward phase, each task is aware of how other tasks are modifying the parameters, making it possible to modify in a more consistent way. Concretely, by communicating with gradients, we can apply consistency constraint to each task during the process of parameter updating, which can prevent private features from slipping into shared space thereby alleviating the *pretend-to-share* problem. Technically, we explore two kinds of communication mechanisms for gradient passing: pairwise and list-wise communication, where the latter one can take task relatedness into consideration.

We evaluate our models on three groups of multi-task learning datasets, range from natural language processing tasks to computer vision. The results show that our models are more expressive since we can learn a shared space which are more pure, driving task-dependent features away.

To summarize, we make the following contributions:

- We describe a general framework, which can not only help us analyze current models in an unified way [Section 3], but find potential limitations, and move forward with suitable approaches [Section 4].
- We study a task-agnostic problem existing in many multi-task learning models, and propose to alleviate it by allowing different tasks to communicate with gradients.
- We present an elaborate qualitative analysis, giving some insight into the internal mechanisms of proposed gradient passing method.

## 2 RELATED WORK

**Multi-task Learning**  There is a large body of works relating to multi-task learning, and we will give systematical analysis in next section. This paper studies a task-agnostic problem existing in many multi-task learning models. Existing models expect to disentangle features into different spaces while features learned in practice are entangled in shared space. Several papers (Liu et al., 2017; Chen et al., 2017) also observe this problem more recently and they propose address it with adversarial training . Despite its effectiveness, it suffers from difficult training process to achieve the balance between generator and discriminator. Besides, adversarial multi-task training methods ignore the relationship between different tasks, leading to low-capacity of learned shared space.

**Gradient Passing Between Samples**  Recently, some works begin to model the dependencies of training samples. Specifically, one sample $x_i$ can pass gradients to another one $x_j$, in which an intermediate function is used to transform gradients into new weights (called fast weights) for $x_j$. This function usually takes different forms, ranging from recurrent neural networks (Andrychowicz et al., 2016; Ravi & Larochelle, 2016), temporal convolutions with attention (Mishra et al., 2018), or memory networks (Santoro et al., 2016). Meanwhile, Finn et al. (2017) explores a special case where the function is an updating function of SGD. However, the samples' dependencies studied by these works are within a task as shown in Fig.1(b), whose motivations are to learn a general updating rule, which can achieve fast adaptation to unseen samples. Different with these works (Finn et al., 2017; Gu et al., 2018), we model the samples' dependencies cross different tasks, aiming to disentangle private and shared features into different spaces.

**Domain Adaptation**  Domain adaptation (DA) alone is a generally well studied area both in natural language processing (Li & Zong, 2008; Blitzer et al., 2007) and computer vision (Saenko et al., 2010) communities. And there are several differences between domain adaptation and multi-task learning (MTL). 1) For domain adaption, there is an explicit pair of source and target domains. Addtionally, the purpose is to adapt the model learned on source domains to target domains. Therefore, DA cares much about the performance on target domains while MTL focuses on performance both in IN-TASK setting and OUT-OF-TASK setting. 2) For domain adaptation, different domains usually have the same label space, and all parameters for different domains are shared, with no need for disentangling

shared and private spaces. More specifically, some recent works introduce meta learning methods for fast domain adaptation (Li et al., 2017; Kang & Feng; Yu et al., 2018). Apart from above two points, the difference with this paper is that our motivation starts from the *pretend-to-share* problem in multi-task learning, then we seek an approach to learn a shared space, which is not contaminated by private features.

## 3 PARAMETERS READ-WRITE NETWORKS

Recent years have seen remarkable success in the use of various neural multi-task learning models to natural languange processing and computer vision. The time is ripe to propose a more general framework, in which we can systematically analyze current models, find potential limitations, and move forward with suitable approaches. With above in mind, in this paper, we propose such a framework, called Parameters Read-Write Networks (PRaWNs), which allow us to dissect existing models and explore additional novel variations within this framework. We will first describe details of PRaWNs.

### 3.1 CONCEPT DEFINITIONS IN PRAWN

Task agents and parameters are two concepts in PRaWN. And the high-level idea of PRaWN to describe multi-task learning process can be expressed as follows: different task agents read information from different types of parameters, and then they write new information to parameters to satisfy some constraints.

**Task Agents** In PRaWN, each task is regarded as an agent with three elements ($\mathcal{D}$, READ-OP, WRITE-OP), in which $\mathcal{D}$ denotes the data belonging to the current task. READ-OP and WRITE-OP represent two operations that the current task can execute. The forms of the tasks can differ in various ways, such as the domain (multi-domain learning) or the language (cross-lingual learning).

**Parameters** Parameters are the objects that operations (READ-OPs and WRITE-OPs) of task agents could execute to. Generally, parameters can be classified into different types based on the following aspects.

- Sharable or Private: Sharable parameters can be read and modified by multiple tasks.

- Readable or Non-readable: If $\theta$ for the task agent $k$ is readable, it means that READ-OP of task $k$ can be executed.

- Writable or Non-writable: If $\theta$ for the task agent $k$ is readable, it means that WRITE-OP of task $k$ can be executed.

| Modes | Properties | | |
|---|---|---|---|
| | Sharable | Writable | Readable |
| $swr$ | ✓ | ✓ | ✓ |
| $\bar{s}wr$ | ✗ | ✓ | ✓ |
| $\bar{s}\bar{w}r$ | ✗ | ✗ | ✓ |
| $\bar{s}\bar{w}\bar{r}$ | ✗ | ✗ | ✗ |

Figure 2: Effective modes of different types parameters for general multi-task learning.

According to above three aspects of parameters, we can enumerate four effective modes for general multi-task learning as shown Fig. 2. We use $swr$ to represent them and the $\bar{r}$ denotes non-readable. For each task, we can understand its sharing schema based on its constituents of parameters. Detailed examples can be found in Appendix.

### 3.2 READ OPERATION

The purpose of READ-OP is to gather useful information $\mathbf{r}^k$ based on current input samples $\mathcal{X}^k$ of task $k$ from all **readable parameters**.

$$\mathbf{r}^k = \text{READ-OP}(\Theta_k^{*r}, \mathcal{X}^k) \tag{1}$$

$$= \text{OUT}(\theta_k^{\bar{s}wr}, \text{ENC}(\mathcal{X}^k, \{\Theta_k^{*r} - \theta_k^{\bar{s}wr}\})) \tag{2}$$

| Read Operations | Sketches | Constituent Para. | Instantiations |
|---|---|---|---|
| Flat READ-OP |  |  | Collobert & Weston (2008); Yang et al. (2016) |
| Star READ-OP |  |  | Yang et al. (2016); Liu et al. (2016; 2017) |
| Structural READ-OP |  |  | Yang & Hospedales (2016); Chen et al. (2018b); Misra et al. (2016) |

Table 1: Typical READ-OPTs with their basic model frameworks, constituent parameters, and related instantiations. The second column illustrates the sketch of each READ-OP, which takes as input two tasks $A$ and $B$. Solid lines show the information flowing of tasks A. Cyan boxes denote private encoders while red boxes represent shared encoders. Blue circles denote output layers. The third column shows the different types of parameters each READ-OP consists of with respect to the task A.

where we refer to $\Theta$ as a set of parameters $\theta$. $\Theta_k^{*r} = \{\Theta_k^{swr}, \Theta_k^{\bar{s}wr}, \Theta_k^{\bar{s}\bar{w}r}\}$ represents all readable parameters with respect to task $k$. OUT denotes output layer while ENC denotes feature encoding layer, ENC can be instantiated as CNN or RNN layer.

For this operation, the key step is how to effectively define readable parameters $\Theta_k^{*r}$, organizing their dependency relationship and then aggregating useful information from them. Specifically, the process to seek for desirable READ-OPs can be abstracted to a **architecture searching** problem (Chen et al., 2018a), which has raised a large body of existing works on this as shown in Tab.1. Here, we show some typical READ-OPs.

**Flat READ-OPs:** $\quad \Theta_k^{*r} = \{\Theta_k^{swr}, \Theta_k^{\bar{s}wr}\}$ and $\{\Theta_k^{\bar{s}wr} - \theta_k^{\bar{s}wr}\} = \varnothing$

Models with flat READ-OP usually assume that all parts of neural network are sharable except output layer , suggesting that different tasks have no private space to encode input samples.

**Star-network READ-OPs:** $\quad \Theta_k^{*r} = \{\Theta_k^{swr}, \Theta_k^{\bar{s}wr}\}$

$\{\Theta_k^{\bar{s}wr} - \theta_k^{\bar{s}wr}\}$ is used for learning task-dependent representations. Different with above models, star-network operation introduces an extra private encoder for each task, aiming to disentangle task-specific and task-invariant features into different spaces.

**Structural READ-OPs:** $\quad \Theta_k^{*r} = \{\Theta_k^{swr}, \Theta_k^{\bar{s}wr}, \Theta_k^{\bar{s}\bar{w}r}\}$

Clearly, for above two operations, there is no definitions of parameters $\theta_k^{\bar{s}\bar{w}r}$ for any $k$, which means that any two tasks can not directly communicate with each other. To address this problem, structural READ-OPs are proposed, where each task can read information from the parameters of other tasks.

### 3.3 WRITE OPERATION

For task agent $k$, the write operation will make modifications to its **writable parameters** to satisfy some constraints.

$$\Theta_k^{*w*} = \text{WRITE-OP}(\Theta_k^w, \Delta_{\Theta_k^{*w*}} \mathcal{L}(\mathbf{r}^k)) \tag{3}$$

where $\Theta_k^{*w*} = \{\Theta_k^{swr}, \Theta_k^{\bar{s}wr}\}$ represents the parameter sets, each of which is writable with respect to task $k$.

During the writing phase, a key question is how to introduce desirable constraints to modify writable parameters $\Theta_k^{*w*}$ for each tasks. Many existing works try to incorporate different kinds of losses to achieve that, where the task-oriented constraint is commonly used to both update $\Theta_k^{swr}$ and $\Theta_k^{\bar{s}wr}$.

**Task-Oriented Constraint**    This constraint usually depends on specific task. For example , for a classification task, $\mathcal{L}_{task}$ can be defined as:

$$\mathcal{L}_{task} = L(\text{READ-OP}, \mathcal{Y}) \tag{4}$$

where $L$ represents the cross-entropy loss.

To incorporate more inductive bias, some extra constraints are introduced. For example, **adversarial constraint** $\mathcal{L}_{Adv}$ is used to encourage shared space more pure while **orthogonal constraint** $\mathcal{L}_{Diff}$ introduce an inductive bias that shared and private feature space should be more orthogonal. More descriptions of these two constraints can be found in Appendix.

## 4    EXPLORING NEW COMMUNICATION PROTOCOLS FOR PRAWN

Given how many specific multi-task learning models have appeared in the literature, an general framework can not only help us deeply understand existing work in an unified way, but can guide us to find potential limitation and motivate us to propose new variations for a specific application.

### 4.1    PROBLEMS AND MOVING FORWARD

In PRaWN framework, a lot of works focus on designing READ-OPs with different forms, aiming to obtain the most suitable sharing schema. However, the *pretend-to-share* problem is less studied in multi-task learning. By dissecting multi-task learning models into different phases within PRaWN, we can get more intuitive understanding towards this problem.

**Parameters Corruption**    We attribute the reason for *pretend-to-share* to the inconsistent WRITE-OPs executed by different tasks. Specifically, each task has the permission to modify shared parameters $\Theta_k^{swr}$ without any communication protocol, which makes it easier to update parameters along diverse directions.

Formally, given two tasks $A$ and $B$, they will both execute WRITE-OP towards their shared parameters $\Theta_k^{swr}$ based on their data $\mathcal{D}^A$ and $\mathcal{D}^B$ as follows:

$$\tilde{\Theta}^{swr} = \text{WRITE-OP}(\Theta^{swr}, \Delta_{\Theta^{swr}}\mathcal{L}(\mathbf{r}^A, \mathcal{D}^A)) \tag{5}$$

$$\tilde{\Theta}^{swr} = \text{WRITE-OP}(\Theta^{swr}, \Delta_{\Theta^{swr}}\mathcal{L}(\mathbf{r}^B, \mathcal{D}^B)) \tag{6}$$

The independency of WRITE-OPs among different tasks make it hard to let multiple tasks update parameters consistently. Therefore, the shared space we finally learned is redundant and contaminated by task-specific features.

### 4.2    GRADIENT PASSING FOR MULTI-TASK COMMUNICATION

With above problems in mind, we propose to introduce a new inductive bias for inter-task communication, in which different tasks should tell others what modifications they have made to shared parameters. Additionally, their communication should support a new protocol that modifications induced by different tasks should be as consistent as possible.

To achieve the above idea, in this paper, we propose two types of approaches as a primary exploration, and hopefully, more effective methods can be put as future work.

**Pair-wise Communication**    In this architecture, we only consider the communication of any two tasks. Specifically, for tasks $A$ and $B$, they can explicitly send their gradient to each other, telling how they modified shared parameters. Simultaneously, the receiver should update parameters with the consideration of the other task's modifications.

Formally, the model will first computed the gradient $\nabla_{\Theta^{swr}} \mathcal{L}^A$ based on task $A$, and then the gradient will be passed to task $B$.

$$\text{READ-OP} = \text{OUT}(\theta_B^{\bar{s}wr}, \text{ENC}(\mathcal{X}^B, \{\Theta_B^{*r} - \theta_B^{\bar{s}wr}\}, \boxed{\nabla_{\Theta^{swr}} L^A})) \tag{7}$$

In practice, the highlighted term can be used in different ways. Here, we utilize them to generate a **fast weight** $\phi^s$ based on $\Theta^{swr}$:

$$\phi^s = g(\Theta^{swr}, \nabla_{\Theta^{swr}} L^A) \tag{8}$$

where $g$ can be a optimizer or parameterized function, which can be defined similar to many existing works relating to meta learning (Andrychowicz et al., 2016; Ravi & Larochelle, 2016; Finn et al., 2017).

Then, our optimized goal is to produce maximal performance on sample $(\mathcal{X}^B, \mathcal{Y}^B)$ using the fast weight $\phi^s$

$$\mathcal{L}_{GP}^{A \leftarrow B} = \min_{\Theta^{swr}} L(\mathcal{D}^B, \phi^s), \tag{9}$$

**List-wise Communication** In this approach, each task $T_k$ will be appended with a task list $\mathcal{T} = \{T_1, \cdots, T_j, \cdots, T_{K-1}\}$ $(T_j \neq T_k)$ and a weight list $\boldsymbol{\beta}_k = \{\beta_1, \cdots, \beta_{K-1}\}$, where $\beta_{kj}$ measures the relatedness between task $k$ and $j$. Moreover, task $T_k$ can communicate with any task from $\mathcal{T}$. Formally, for task $k$, we pass its gradients to the other tasks and achieve maximal performance on corresponding samples $D^j$.

$$\mathcal{L}_{GP}^{k \leftarrow \cdot} = \sum_{j \sim \mathcal{T}} \min_{\Theta^{swr}} \beta^{k \leftarrow j} \mathcal{L}^{k \leftarrow j}(\mathcal{D}^j, \phi^s), \tag{10}$$

where $\mathcal{D}^j$ denotes training samples from task $j$, $\beta^{k \leftarrow j}$ describes the task relationship between $k$ and $j$, which can be pre-defined empirically or learned dynamically.

**Require:** A set of training tasks $\{\mathcal{D}^{\mathcal{T}_i}\}_{i=1}^K$, where $\mathcal{T}_i$ is drawn from $p(\mathcal{T})$
1: Initialize $\Theta := \{\theta^s, \theta_1^p, \cdots, \theta_K^p\}$
2: **while** not done **do**
3:     **for** $\mathcal{T}_k \sim p(\mathcal{T})$ **do**
4:         Sample a batch of samples $\mathcal{B}^{\mathcal{T}_k} = \{X, Y\} \in \mathcal{D}^{\mathcal{T}_k}$
5:         Evaluate gradients $\nabla_{\theta^s, \theta_k^p} \mathcal{L}_{task}(\theta^s, \theta_k^p, \mathcal{B}^{\mathcal{T}_k})$
6:         Update parameters: $\theta_k^p \leftarrow \alpha \nabla_{\theta_k^p} \mathcal{L}_{task}$
7:         **if** `Pairwise-Communication` **then**
8:           Sample a batch $\mathcal{B}^{\mathcal{T}_j} = \{X, Y\} \in \mathcal{D}^{\mathcal{T}_j}$ $\triangleright \mathcal{T}_k \neq \mathcal{T}_j$
9:           $\mathcal{L}_{GP} = \min_{\theta^s} \mathcal{L}(\mathcal{B}^{\mathcal{T}_j}, \boxed{\nabla_{\theta^s} \mathcal{L}^k}, \theta^s)$
10:        **else if** `Listwise-Communication` **then**
11:          Compute a weight list $\boldsymbol{\beta}_k$
12:          **for** $\mathcal{T}_j \sim p(\mathcal{T})$ **do** $\triangleright \mathcal{T}_k \neq \mathcal{T}_j$
13:           $\mathcal{L}_{GP}^{k \leftarrow j} = \min_{\theta^s} \mathcal{L}(\mathcal{B}^{\mathcal{T}_j}, \boxed{\nabla_{\theta^s} \mathcal{L}^k}, \theta^s)$
14:          **end for**
15:          $\mathcal{L}_{GP} = \sum_{j \sim \mathcal{T}} \min_{\theta^s} \beta^{k \leftarrow j} \mathcal{L}_{GP}^{k \leftarrow j}$
16:        **end if**
17:        Update parameters: $\theta^s \leftarrow \alpha \nabla_{\theta^s} \mathcal{L}_{GP}$
18:     **end for**
19: **end while**

Figure 3: Gradient Passing for Multi-task Communication. The superscript $s$ and $p$ represent parameters which can be shared or not (private).

**Empirical Methods for Obtaining Weight Lists** We provide a simple way to compute $\boldsymbol{\beta}$ for these tasks which have similar output spaces, such as multiple domains or languages. we take turns choosing one task and use its training data to train a basic model (will be described as follows.). Then, we use the testing data of the remaining tasks to test the basic model. In this way, we can obtain a matrix $\mathbf{V} \in \mathcal{R}^{K \times K}$, in which each cell $\mathbf{V}_{i,j}$ stores the accuracy of testing task $j$, which is evaluated on model learned from training task $i$. We then update matrix $\mathbf{V}$ as following operation: for each column $j$:

$$\tilde{\mathbf{V}}_{i,j} = \begin{cases} 1 & i = j \\ v^{\frac{1}{q}} & \text{otherwise} \end{cases} \tag{11}$$

where $v = min\{\mathbf{V}_{i,j}/\mathbf{V}_{j,j}, 1\}$ and $q$ is a hyper-parameter. Then $\tilde{\mathbf{V}}_{i,j}$ can be used as a weight $\beta_{kj}$.

### 4.3 PUT IT ALL TOGETHER

We put task-oriented constraint $\mathcal{L}_{task}$ and gradient passing $\mathcal{L}_{GP}$ together, which can both learn private features and learn more universal shared rules, since explicitly gradient passing allow shared parameters to be updated in a more consistent way.

## 5 EXPERIMENT

In this section, we investigate the empirical performance of our proposed models on three groups of multi-task learning datasets: text classification (Multi-Sent) (Liu et al., 2017), image aesthetic assessment (Murray et al., 2012) (Multi-AVA), and sequence tagging. Each dataset contains several related tasks. The reason that we utilize these datasets is they can provide a sufficient testing ground (i.e. IN-TASK setting and OUT-OF-TASK setting).

### 5.1 TASKS AND DATASETS

In this section, we will give a brief description about evaluated tasks and datasets.

**Text Classification** This dataset contains different text classification tasks involving different popular review corpora, such as "`books`" and "`apparel`" Liu et al. (2017). Each sub-task aims at predicting a correct sentiment label (positive or negative) for a given sentence. All the datasets in each task are partitioned into a training set, development set, and test set with the proportions of 1400, 200, and 400 samples respectively.

**Sequence Labelling** We choose POS, Chunking, and NER as evaluation tasks on Penn Treebank, CoNLL 2000, and CoNLL 2003 respectively.

**Image Aesthetic Assessment** The image aesthetic assessment can be formulated as a classification problem of predicting an image high- or low-aesthetic categories within the supervised learning framework (Murray et al., 2012; Deng et al., 2017). The multi-domain AVA dataset contains 90,000 images covering 10 different domains For each domain, the data is split into training (6,000), validation (1,000), and testing (2,000) sets.

### 5.2 EXPERIMENTAL SETTINGS AND IMPLEMENTATION DETAILS

For Multi-Sent and Multi-AVA datasets, we randomly divide 10 tasks into two groups for IN-TASK setting and OUT-OF-TASK setting. As a contrast, each task will first be evaluated separately by corresponding base models. For text classification and sequence labelling tasks, the base models are implemented as (Liu et al., 2017) (LSTM encoder with MLP layer) and (Huang et al., 2015) (LSTM encoder with CRF layer). For image aesthetic assessment task, we adopt a 50-layer Residual Network (He et al., 2016) as the base model. Specifically, for an image $I$, it will be first vectorized into a feature vector $\mathbf{x}$ by the penultimate layer of residual network, which has been pre-trained on Imagenet. Then the obtained feature vector $\mathbf{x}$ will be passed into two-layer fully-connected layers, which followed by a softmax function.

**Model Settings** We evaluate models under two specific evaluation settings, including IN-TASK, OUT-OF-TASK. For OUT-OF-TASK setting, learned shared parameters will be combined with new parameters for a task, which has not be seen during training phase in IN-TASK setting. Then all the parameters will be learned with a few training samples of new task.

Besides, we select different combinations of READ-OPs and constraints of WRITE-OPs as basic multi-task learning frameworks, then investigate their effectiveness when gradient passing mechanism is introduced or not, which results in the following evaluated models:

- **FR**: Flat READ-OP with task-specific constraint.
- **SR**: Star-network READ-OP with task-specific constraint.
- **ASR**: SR with adversarial training and orthogonal constraints.

Our models are built upon FR and SR, and we use the prefixes "PGP" and "LGP" to denote pair-wise gradient passing and list-wise gradient passing respectively.

### 5.3 MAIN RESULTS

**In-task Setting** Tab.2 shows performances of our models on three groups of datasets. For in-task setting, we can obtain the following observations: 1) For different types of tasks, READ-OPs perform

| Tasks | | Single Task | Multi-task Framework | | | | | | |
|---|---|---|---|---|---|---|---|---|---|
| | | Base Models | FR | SR | ASR | PGP-FR | PGP-SR | LGP-FR | LGP-SR |
| **Multi-Sent** | | | | | IN-TASK SETTING | | | | |
| | Books | 79.5 | Δ0.5 | Δ0.2 | Δ2.0 | Δ3.5 | Δ5.0 | Δ4.2 | Δ3.7 |
| | Elect. | 80.5 | Δ1.7 | Δ2.7 | Δ2.7 | Δ2.2 | Δ4.5 | Δ5.0 | Δ5.0 |
| | DVD | 81.7 | Δ0.3 | −Δ1.2 | Δ2.5 | Δ2.0 | Δ2.1 | Δ0.3 | Δ4.0 |
| | Kitchen | 78.0 | Δ3.7 | Δ7.5 | Δ8.0 | Δ7.5 | Δ7.5 | Δ7.5 | Δ8.5 |
| | Apparel | 83.2 | Δ1.0 | Δ0.3 | Δ1.3 | Δ0.8 | Δ3.0 | Δ0.1 | Δ2.8 |
| | Average | | Δ1.4 | Δ1.9 | Δ3.3 | Δ3.2 | Δ4.4 | Δ3.4 | **Δ4.8** |
| | | | | | OUT-OF-TASK SETTING | | | | |
| | Camera | 85.2 | 78.5 | 78.3 | 81.8 | 80.3 | 80.7 | **82.5** | 81.7 |
| | Health | 85.5 | 78.5 | 78.5 | 79.3 | 79.7 | **81.0** | 80.5 | 80.3 |
| | Music | 77.7 | 70.8 | 74.8 | 72.5 | 75.0 | **76.0** | 75.7 | 76.5 |
| | Toys | 83.2 | 81.2 | 80.5 | 81.0 | **82.5** | 81.5 | 80.7 | 81.5 |
| | Video | 81.5 | 73.3 | 77.7 | 77.0 | 77.0 | **79.3** | 78.5 | 78.8 |
| **Multi-AVA** | | | | | IN-TASK SETTING | | | | |
| | Abstract | 66.6 | Δ2.1 | - | - | Δ5.3 | - | Δ4.6 | - |
| | Animals | 70.5 | -Δ0.1 | - | - | Δ2.2 | - | Δ1.8 | - |
| | Black | 70.6 | -Δ2.7 | - | - | Δ0.0 | - | -Δ0.6 | - |
| | Macro | 68.4 | Δ0.7 | - | - | Δ2.9 | - | Δ3.0 | - |
| | Still | 65.4 | Δ2.1 | - | - | Δ3.6 | - | Δ3.8 | - |
| | Average | 68.3 | Δ0.4 | - | - | Δ2.8 | - | **Δ2.5** | - |
| | | | | | OUT-OF-TASK SETTING | | | | |
| | Emotive | 70.1 | 65.9 | - | - | 63.9 | - | **67.1** | - |
| | Humor | 66.1 | 58.8 | - | - | **65.5** | - | 61.7 | - |
| | Lands. | 69.2 | 56.4 | - | - | 57.5 | - | **60.4** | - |
| | Nature | 65.5 | 56.8 | - | - | **62.9** | - | 60.2 | - |
| | Portrait | 68.7 | 62.4 | - | - | 61.8 | - | **64.6** | - |
| **Tagging** | Chunking | 94.46 | 94.21 | 95.41 | 95.27 | 95.02 | 95.55 | 95.20 | **95.66** |
| | NER | 90.10 | 90.54 | 90.94 | 90.90 | 90.72 | **91.57** | 91.25 | 91.42 |
| | POS | 97.55 | 97.47 | 97.55 | 97.61 | 97.57 | 97.75 | 97.62 | **97.77** |

Table 2: Performances of our models on three groups of multi-task learning datasets against typical baselines. Δ denotes the improvement value compared with the base model under single task setting. In OUT-OF-TASK setting, we transfer shared layers learned during in-task setting to new tasks, which are not seen during in-task setting. For `Multi-AVA` dataset, there is no model with star-network READ-OP, since here the feature extractor (ResNet) is used as a pre-trained encoder, whose parameters are fixed and we don't learn a separate ResNet from scratch.

diversely. For example, on `Multi-Sent` dataset, both flat and star-network READ-OP can achieve significant improvements compared with single task model. However, for `tagging` task, FR model (flat READ-OP ) works worse than single task model while the performance will be improved when SR is utilized. We attribute the reason to the higher discrepancy existing in tagging tasks. (The output space of each task in `tagging` is totally different.) 2) The performances of FR and SR can be significantly improved when gradient passing mechanism is introduced. Specifically, LGP-SR outperforms other models on `Multi-Sent` dataset, which not only shows the effectiveness of propose methods but indicates the importance of taking relatedness of tasks into consideration. 3) PGP-SR and LGP-SR can surpass ASP both on `Multi-Sent` and `tagging` datasets, which suggests that it's more effective for multi-task learning to allow different tasks to communicate by gradients.

**Out-of-Task Setting** This setting is designed to test the transferability of shared knowledge for different models. As shown in Tab.2, on `Multi-Sent` dataset, our model PGP-SR and LGP-SR can achieve comparable results with single task model, which is trained on 1600 samples, while we just use 200 samples. Additionally, on both `Multi-AVA` and `Multi-Sent` datasets, READ-OPs

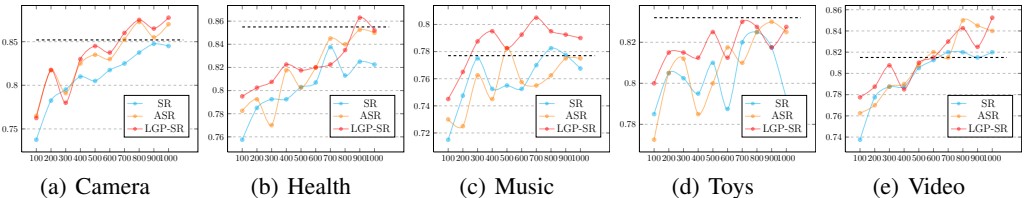

(a) Camera    (b) Health    (c) Music    (d) Toys    (e) Video

Figure 4: Performances of SR (Star-network), ASR (Adversarial Star-network) and LGP-SR (Star-network with list-wise gradient passing) models with different numbers (from 100 to 1000) of training samples on five tasks. The darker grey horizontal line shows the performances in single-task setting (with 1600 training samples).

with gradient passing mechanisms show better generalization to unseen tasks, which further suggests that our models can capture more universal features during the training phase in the IN-TASK setting.

## 5.4 ANALYSIS

### 5.4.1 IMPACT OF THE DIFFERENT NUMBERS OF TRAINING SAMPLES

For OUT-OF-TASK setting, we additionally investigate how the performances changes for different models as the number of training samples (#samples) changes. Specifically, here we plotted the accuracy curves of our model (LGP-SR) against two baselines SR and ASR on five text classification tasks. As shown in Fig.4, we can get following observations: 1) When #samples is less than 500, LGP-SR can achieve the fastest adaptation on last four tasks. 2) For each task, the highest performances are achieved by LGP-SR. Additionally, for "Camera,Health,Music,Video" tasks, the performance of MPS can surpass the model in single-task setting, where 1600 training samples are used. Notably, for "Camera, Music, Video" tasks, performances of LGP-SR can reach the grey horizontal line using half the training data.

### 5.4.2 EVOLUTION OF PARAMETERS

To get more intuitive understanding of how gradient passing mechanism influences the shared spaces, we visualize how the shared and private parameters' values evolve under our model PGP-SR compared to the typical baseline SR. Specifically, we randomly chose two tasks "electronics" and "books" to train models SR and PGP-SR. Then, during the training phase, we projected the parameter space using PCA and plot their curves as the epoch increases. As shown in Fig.5, we can observe that: for SR model, when the updating directions of two private parameters are inconsistent, shared parameters will be updated along one of them. However, for PGP-SR as shown in Fig.5-(b), the updating direction of shared parameters is an integration of two private updating directions, which can prevent shared parameters from fitting more task-specific features.

### 5.4.3 BEHAVIOURS OF NEURONS IN SHARED LAYER

We can gain some insight into the internal mechanisms of gradient passing method by studying the neuron activations in shared layer $\mathbf{h} \in \mathcal{R}^d$ as they process test set data of different tasks ($d$ is the number of neurons). We were particularly interested in looking at which parts of words can be regraded as useful features by neurons in shared layer. To achieve this, we introduce a variable $q$ for each word $w$, which is used to describe what the possibility is that the word can be regarded as a useful feature by neurons in $\mathbf{h}$. Formally, we refer to $n_w$ as the times that $w$ appears in test data and $d$ is the number of neurons. Then, we define $q = \frac{N_{max}}{(n_w \times d)}$, where $N_{max}$ denotes how many times neurons achieve the highest activation on the word $w$. We compute $q$ for each word in $V$ with SR and LGP-SR models, in which $V = \{V_1, \cdots, V_5\}$ denotes the vocabulary of five test sets (in-task). We then divide $V$ into two parts: the intersection of five sub-vocabularies $\bigcap_i^5 V_i$ and its

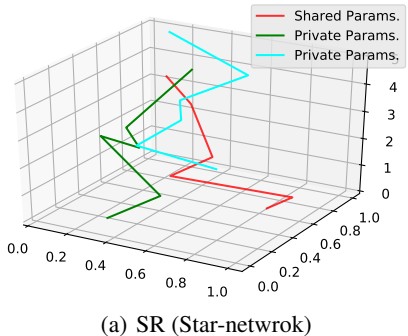 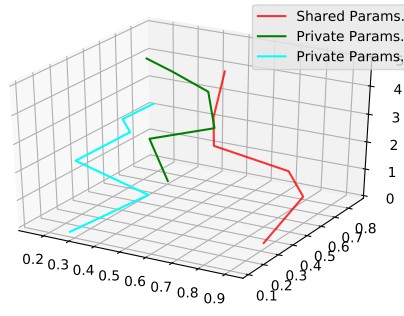

(a) SR (Star-netwrok)  (b) PGP-SR (Pair-wise Gradient Passing)

Figure 5: Evolution of parameters during training phase projected into 3D space using PCA for SR and PGP-SR models. $z$-axis describes different epochs. Cyan and green lines represent the evolution of private parameters in task "`electronics`" and "`books`", while the red line denotes the evolution of parameters shared by these two tasks.

| Settings | | Features |
|---|---|---|
| $\bigcap_i^5 V_i$ | SR | definitely, excellent, gift, doesnt, fantastic, wonderfully, awesome, pleased, perfectly,alright |
| | LGP-SR | disappointed, doesnt, reliable, sucks, excellent, pleased, regret, fantastic, perfectly, amazing |
| $\overline{\bigcap_i^5 V_i}$ | SR | Brosnan, boils, mixes, Lasko, sauce, straightened, breathable, distinctly, havent, sweatpants |
| | LGP-SR | richness, dissatisfied, havent, roaster, overlong, doesnt, meaty, easiest, orginally, buzzes |

Table 3: Top-10 word features in share space with high values of $q$ on two parts of vocabularies. The shared spaces are constructed by SR and LGP-SR models respectively. $\bigcap_i^K 5_i$ denotes the intersection of five sub-vocabularies and $\overline{\bigcap_i^5 V_i}$ represents its complement.

complement $\overline{\bigcap_i^5 V_i}$, which allows us to easily analyze the characteristics of learned features in shared space between two models [1].

Tab.3 shows the top 10 word features with high values of $q$ on two parts of vocabularies and we can obtain the following observations: 1) Both SR and LGP-SR can easily pick up useful and sharable words in $\bigcap_i^5 V_i$. According to our statistics, there are 1073 words in $\bigcap_i^5 V_i$, and we find the words with high values of $p$ do make sense. For examples, the words with higher ranking "`excellent, disappointed, doesnt`" are key patterns for all tasks to make prediction of one sentence's sentiment. 2) For words in $\overline{\bigcap_i^5 V_i}$, the SR model mistakenly regards more words involving task-specific information as sharable features, such as the name "`Brosnan`" and entity "`sauce`". By contrast, LGP-SR still can pick up useful words such as "`dissatisfied, havent, overlong`".

## 6 CONCLUSION

In this paper, we introduce an inductive bias for multi-task learning, allowing different tasks to pass gradients explicitly, which makes it available to restrict different tasks make consistent parameter updating. In this way, we can avoid the leakage of private information into the shared space The empirical results not only show the effectiveness of our proposed method, but give some insight into its internal mechanisms.

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

## 7 APPENDIX

### 7.1 EXTRA CONSTRAINTS FOR WRITE-OPS

**Adversarial Constraint** This loss is used to encourage shared space more pure,

$$\mathcal{L}_{Adv} = \min_{\{\Theta^{swr} - \theta^{swr}\}} \left( \max_{\theta^{swr}} L(\text{OUT}(\theta^{swr}, \text{ENC}(\mathcal{Y}^k, \{\Theta^{swr} - \theta^{swr}\}))) \right) \quad (12)$$

where $\mathcal{Y}^k$ represent the true labels for the types of current tasks.

**Orthogonal Constraint** This constraint introduce an inductive bias that shared and private feature space should be more orthogonal.

$$\mathcal{L}_{Diff} = \left\| \mathbf{S}^{k^\top} \mathbf{P}^k \right\|_F^2 , \quad (13)$$

where $\| \cdot \|_F^2$ is the squared Frobenius norm. $\mathbf{S}^k$ and $\mathbf{P}^k$ are two matrics, whose rows are the output of shared extractor $\text{OUT}(\mathcal{X}^k, \Theta_k^{swr})$ and task-specific extractor $\text{OUT}(\mathcal{X}^k, \Theta_k^{\bar{s}wr})$ of an input sentence.

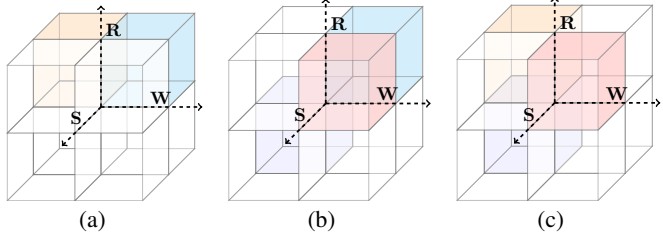

Figure 6: (a) Single task Models. (b) Hard-sharing Multi-task Models. (c) Soft-sharing Multi-task Models.

## 7.2 PARAMETER DESCRIPTIONS WITH OCTANT

Intuitively, for parameters, different cases of modes can be described in an octant. And Fig. 6(a-c) illustrate three typical learning models: single task models, mulit-task model with hard-sharing and multi-task model with soft-sharing.

## 7.3 TRAINING AND HYPER-PARAMETERS

**Image Aesthetic Assessment**  On this task, we randomly crop out a 224 x 224 window to feed into the ResNet-50, which is pre-trained on ImageNet dataset. We utilize the SGD, Adam to optimize the parameters with the fixed learning rate 0.001 for the classification task and regression task, respectively. The mini-batch size is 96 and the number of iterations is 128. The Weight list $\beta_k$ for Multi-AVA can be seen in Tab.4.

**Text Classification and Tagging**  The word embeddings for all of the models are initialized with the 200-dimensional GloVe vectors (840B token version (Pennington et al., 2014)). The other parameters are initialized by randomly sampling from the uniform distribution of $[-0.1, 0.1]$. The mini-batch size is set to 8. We apply stochastic gradient descent with the diagonal variant of AdaDelta for optimization (Zeiler, 2012). The Weight list $\beta_k$ for Multi-Sent can be seen in Tab.5. For tagging task, we regard each task equally.

|          | Abstract | Black | Still | Macro | Animals |
|----------|----------|-------|-------|-------|---------|
| Abstract | 1        | 0.86  | 0.91  | 0.98  | 0.85    |
| Black    | 0.81     | 1     | 0.79  | 0.80  | 0.78    |
| Still    | 1        | 0.89  | 1     | 0.98  | 0.80    |
| Macro    | 0.98     | 0.79  | 0.92  | 1     | 0.79    |
| Animals  | 0.91     | 0.94  | 0.96  | 0.95  | 1       |

Table 4: Weight list $\beta_k$ for Multi-AVA.

|         | Books | Elec. | DVD  | Kitchen | Books |
|---------|-------|-------|------|---------|-------|
| Books   | 1     | 0.73  | 0.95 | 0.75    | 0.77  |
| Elec.   | 0.80  | 1     | 0.79 | 0.92    | 0.90  |
| DVD     | 0.98  | 0.78  | 1    | 0.76    | 0.80  |
| Kitchen | 0.94  | 0.99  | 0.73 | 1       | 0.95  |
| Books   | 0.79  | 0.94  | 0.74 | 0.91    | 1     |

Table 5: Weight list $\beta_k$ for Multi-Sent.

