# OpenReview forum: "Multi-task Learning with Gradient Communication"
_ICLR.cc/2019/Conference_

### Official Review · AnonReviewer1 · 2018-10-30

**Rating:** 7
**Confidence:** 3

**Review:**

This paper proposes that models for different tasks in multi-task learning cannot only share hidden variables but also gradients.

Pros:
- The overall framework is theoretically motivated and intuitive. The idea of passing gradients for multi-task learning is interesting and the execution using fast weights is plausible.
- The experiments are extensive and cover three different task combinations in different domains.
- The results are convincing and the additional analyses are compelling.

Cons:
- I would have liked to see a toy example or at least a bit more justification for the "pretend-to-share" problem that models "collect all the features together into a common space, instead of learning shared rules across different tasks". As it is, evidence for this seems to be mostly anecdotal, even though this forms the central thesis of the paper.
- I found the use of Read and Write ops confusing, as similar terminology is widely used in memory-based networks (e.g. [1]). I would have preferred something that makes it clearer that updates are constrained in some way as "writing" implies that the location is constrained, rather than the update minimizing a loss.

Questions:
- How is the weight list of task similarities \beta learned when the tasks don't share the same output space? How useful is the \beta?
- Could you elaborate on what is the difference between pair-wise gradient passing (PGP) and list-wise gradient passing (LGP)

[1] Graves, A., Wayne, G., & Danihelka, I. (2014). Neural turing machines. arXiv preprint arXiv:1410.5401.

---

> ### Author Response · Authors · 2018-11-26
> **Response to reviewer 1**
>
> Thanks for your comments and the response of each point is listed below.
> That’s a really good question. As described in our paper, for the tagging tasks(POS, Chunking,NER), they don’t share the same output space, and here we regard each task equally. Another alternative way is to learn them dynamically.
> Both pairwise and listwise communication mechanisms are designed for addressing the inconsistent updating problem of shared parameters between different tasks.
> The difference is that pairwise communication considers the updating consistency of parameter between two tasks, which is a relatively relaxed constraint.
> Specifically, the constraint of listwise communication is used to purify the intersection of all task’s feature space while pairwise communication is more fit to the partially-shared scenario.

---

### Official Review · AnonReviewer3 · 2018-11-01
**just packing exsiting algorithms**

**Rating:** 4
**Confidence:** 4

**Review:**

This paper tries to address the"pretend-to-share" problem by designing the gradient passing schemes in which the gradient updates to specific parameters of tasks are passed to the shared parameters. Besides, the authors summarize existing multitask learning algorithms in a framework called Parameters Read-Write Networks (PRAWN).

Pros:
- The view of putting existing multi-task learning algorithms in a read-write framework is quite intriguing and inspiring.

Cons:
- Motivation: The whole paper is assumed to address the "pretend-to-share" problem, while the authors never provide any evidence that such problem really exists for any other algorithm. It seems to be an assumption without any support.
- Method:
   - Though the read-write framework is very interesting, the authors do not clearly present it, so that the readers can be totally get lost. For example, what do you mean by writing {\Theta^{*r}_k - \theta^{swr}_k}? In the line of structural read-op, where are \theta_3 and \theta_4  in the column of the constituent para. ? What do you mean by writing the equation (4)? How do you define g() in equation (8)? This is a research paper which should be as clear as possible for the readers to reproduce the results, rather than a proposal only with such abstract and general functions defined.
   - In the list-wise communication scheme, you define the task relationship in equation (11). The problem is how do you evaluate the effectiveness of such definition, since massive works in multitask learning pursue to learn the task relationship automatically to guarantee the effectiveness instead of such heuristic definition.
- Related works: The authors do not clearly and correctly illustrate the connections between this work and meta-learning/domain adaptation. To my best knowledge, meta-learning, including MAML (Finn et al. 2017), can obviously solve both in-task setting and out-task setting. In some sense, I think this work is almost equivalent to MAML.
- Experiments:
   - First, several state-of-the-art baselines including MAML and cross-stitch networks should be compared. Specifically, for the text classification dataset, there have been a lot of domain adaptation works discovering the transferable pivots (shared features) and non-pivots (specific features), which the authors should be aware of and compare in Table 3.
   - The Figure 5 is not clear to me, and so is the discussion. The authors try to explain that the updating direction of shared parameters for PGP-SR is an integration of two private updating directions. I tried hard to understand, but still think that Figure 5(a) is even better than Figure 5(b). The updating direction of the shared parameters is almost the same as the cyan line.
- Presentation: there are so many grammatical errors and typos. For example,
   - In the introduction, "...datasets, range from natural" -> "...datasets, ranging from natural"
  - In the related work, "and they propose address it with adversarial" -> "and they propose to address it with adversarial"
 - In the beginning of Section 4, " an general" -> "a general"

---

> ### Author Response · Authors · 2018-11-26
> **Response to reviewer 3**
>
> We thanks for your insightful and helpful comments.
> First, we have made our presentation more clear based on your confusion:
> 1) Eq.(4) (8)
> 2) Description of Figure (5)
> Some detailed responses are shown as follows:
>
> Q1: “The whole paper is assumed to address the "pretend-to-share" problem, while the authors never provide any evidence that such problem really exists for any other algorithm. It seems to be an assumption without any support.”
> A1: Actually the term “pretend-to-share” is just used to describe an existing problem and related evidence has been observed in some previous work (Bousmalis et al., 2016; Liu et al., 2017).  While we have claimed this in “Introduction” section, we will make this more clear in our revised version.
>
>
>
> Q2: “To my best knowledge, meta-learning, including MAML (Finn et al. 2017), can obviously solve both in-task setting and out-task setting. In some sense, I think this work is almost equivalent to MAML”
> A2:
> 1) First of all, we don’t claim that meta-learning methods cannot be used for in-task setting and out-of-task setting. The thing we claimed is existing methods about meta learning focus on modelling the dependencies of samples from the SAME tasks.
> 2) Most existing meta learning methods are designed for few-shot learning (including MAML), whose motivation is far different from multi-task learning. For multi-task learning, one of the questions we care about is “how to allow different tasks to help each other as more as possible”.
> 3) In this paper, one major contribution is that we find an existing problem in multi-task learning can be alleviated by allowing different tasks to pass gradients.
> To summarize, this paper starts from a real problem existing in multi-task learning, and we propose to solve it by passing gradients between different tasks. Similarly, most existing meta learning methods propose to passing gradient within the same tasks for few-shot learning scenario.
>
> Q3: “several state-of-the-art baselines including MAML and cross-stitch networks should be compared”
> A3: As we have described above, MAML has its own training setting(training set, support set, test set), which is hard to use for our tasks.
> Additionally, there are too many existing methods for multi-task learning, we have chosen existing models as our baselines which also focus on “pretend-to-share” problem.

---

### Official Review · AnonReviewer2 · 2018-11-02
**interesting paper on improving MTL using gradient communication**

**Rating:** 5
**Confidence:** 4

**Review:**

Paper summary:
In this paper, the authors propose a general framework for multi-task learning (MTL) in neural models. The framework is general for including some of the current neural models for MTL. Under the framework, the author propose a new method that could allow tasks to communicate each other with explicit gradients. Based on the gradients being communicated, the system could adjust the updates of one task based on the gradient information of the other task. Also, prior task relatedness information could be incorporated to the system.

The idea of incorporating passing gradients among tasks seems very interesting, which is new as far as I am aware of. Although the idea is simple, but it seems intuitive since purely aggregating gradient updates might have undesired cancelling effects on each other.

There are some questions I have about this method.
1.	I’m curious about how the sequential update in pairwise task communication affects the performance.
2.	Also, how does sequential update nature of the method affect the training speed, as for now, the parameter update consists of two sequential steps which also involve changes to the traditional update rule.
3.	What is fast weight for and how it is used in (9)? It would be better if there are more details on how the update is carried out during the gradient communication.
4.	Regarding the relatedness for List-wise communication, is it possible to update the relatedness dynamically? Since the pre-computed relatedness might not always make sense. During the learning of the representations, the task relatedness could change in the process.
The system framework for MTL introduced by the authors seem to be kind of isolated to the method proposed. I feel that the framework is not quite easy to understand from the way it is presented.  From my perspective, the effectiveness of analyzing MTL methods using the framework seems a bit limited to me, as it serves more like a way of abstracting MTL models instead of analyzing it. Therefore, I feel the content devoted to that part might be too much.

Overall, I think the paper is interesting although the method itself is relatively simple. And the direction of utilizing gradient communication among tasks seem interesting and could be further explored. But I do feel the organization of the paper is a bit too heavy on the framework instead of the methodology proposed. And more details of the algorithm proposed could be provided.

On a side note, I think the paper exceeds the required length limit of 10 pages if appendices are counted towards it.

---

> ### Author Response · Authors · 2018-11-26
> **Response to reviewer 2**
>
> Thanks for your comments and the response of each point is listed below.
>       1.  Both pairwise and listwise communication mechanisms are designed for addressing the inconsistent updating problem of shared parameters between different tasks. The difference is that pairwise communication considers the updating consistency of parameter between two tasks, which is a relatively relaxed constraint. (In the real scenario, there are features which can be shared partially).
>       2.  Yes, explicitly passing gradients to different tasks will take additional time while the overall training processing is still very efficient.
>       3.  Here we choose a function without learnable parameters to compute the fast weight. We have give more detailed formulation in our revised version.
>       4. Yes, as we have also claimed in the paper, the relatedness can be computed in a static or dynamic way. The question “how to choose weights for different tasks ?” is a classic problem and pre-compute the task relatedness has been widely used in existing work. Here, we don’t explore more about this to make our paper more focused.

---

### Meta-Review · Area_Chair1 · 2018-12-13
**Interesting MTL approach, but the work could be improved in the light of suggestions.**

**Confidence:** 4
**Recommendation:** Reject

**Metareview:**

This paper presents a novel idea of transferring gradients between tasks to improve multi-task learning in neural network models. The write-up includes experiments with multi-task experiments with text classification and sequence labeling, as well as multi-domain experiments. After the reviews, there are still some open questions in the reviewer comments, hence the reviewer decisions were not updated.
For example, the impact of sequential update in pairwise task communication on performance can be analyzed. Two reviewers question task relatedness and the impact of how and when it is computed could be good to include in the work. Baselines could be improved to reflect reviewer suggestions.